2HybridTools, a handy software to facilitate clone identification and mutation mapping from yeast two-hybrid screening

Cauchy Pierre cauchy@ie-freiburg.mpg.de 1 2 3 4 5
Kahn-Perlès Brigitte 3 4 5
Ferrier Pierre 2 5
Imbert Jean 3 4 5
Lécine Patrick patrick.lecine@bioaster.org 4 5 6
1 Max Planck Institute for Immunobiology and Epigenetics , Freiburg , Germany
2 Centre d’Immunologie de Marseille-Luminy, Inserm U1104, CNRS UMR7280 , Marseille , France
3 TAGC, Inserm U1090 , Marseille , France
4 Centre de Recherche en Cancérologie de Marseille, Inserm UMR1068, CNRS UMR7258 , Marseille , France
5 Université de la Mediterranée (Aix-Marseille II) , Marseille , France
6 Vaccine Thematic Unit, BIOASTER , Lyon , France
Dinesh-Kumar Savithramma
Electronic publication date: 2019 Jul 3
Publication date: 2019
Volume: 7
Electronic Location ID: e7245
Received 2019 Jan 9; Accepted 2019 Jun 2
Copyright: ©2019 Cauchy et al.
Copyright year: 2019
Copyright holder: Cauchy et al.
License: This is an open access article distributed under the terms of the Creative Commons Attribution License, which permits unrestricted use, distribution, reproduction and adaptation in any medium and for any purpose provided that it is properly attributed. For attribution, the original author(s), title, publication source (PeerJ) and either DOI or URL of the article must be cited.
License URL: https://creativecommons.org/licenses/by/4.0/

Keywords: Two-hybrid, Reverse two-hybrid, Analysis, Software, Identification, Mutation

Funding: Institut National du Cancer (INCa) Fondation pour la Recherche Medicale (FRM) Ligue Nationale contre le Cancer (LNCC) Association pour la Recherche contre le Cancer (ARC) Agence Nationale pour la Recherche (ANR) GIS IBiSA (Infrastructures en Biologie Santé et Agronomie) Institutional grants from Inserm and CNRS Commission of the European Communities The “Fondation de France/Comité Leucémie” The “Fondation Princesse Grace de Monaco” The “Fondation Laurette Fugain” This work was supported by Institut National du Cancer (INCa), Fondation pour la Recherche Medicale (FRM), Ligue Nationale contre le Cancer (LNCC), Association pour la Recherche contre le Cancer (ARC), Agence Nationale pour la Recherche (ANR), GIS IBiSA (Infrastructures en Biologie Santé et Agronomie), institutional grants from Inserm and CNRS, and by specific grants from the Commission of the European Communities, the “Fondation de France/Comité Leucémie”, the “Fondation Princesse Grace de Monaco”, and the “Fondation Laurette Fugain”. The funders had no role in study design, data collection and analysis, decision to publish, or preparation of the manuscript.

==============================
Yeast Two-Hybrid (Y2H) and reverse Two-Hybrid (RY2H) are powerful protein–protein interaction screening methods that rely on the interaction of bait and prey proteins fused to DNA binding (DB) and activation domains (AD), respectively. Y2H allows identification of protein interaction partners using screening libraries, while RY2H is used to determine residues critical to a given protein–protein interaction by exploiting site-directed mutagenesis. Currently, both these techniques still rely on sequencing of positive clones using conventional Sanger sequencing. For Y2H, a screen can yield several positives; the identification of such clones is further complicated by the fact that sequencing products usually contain vector sequence. For RY2H, obtaining a complete sequence is required to identify the full range of residues involved in protein–protein interactions. However, with Sanger sequencing limited to 500–800 nucleotides, sequencing is usually carried from both ends for clones greater than this length. Analysis of such RY2H data thus requires assembly of sequencing products combined with trimming of vector sequences and of low-quality bases at the beginning and ends of sequencing products. Further, RY2H analysis requires collation of mutations that abrogate a DB/AD interaction. Here, we present 2HybridTools, a Java program with a user-friendly interface that allows addressing all these issues inherent to both Y2H and RY2H. Specifically, for Y2H, 2HybridTools enables automated identification of positive clones, while for RY2H, 2HybridTools provides detailed mutation reports as a basis for further investigation of given protein–protein interactions.

Introduction

Determining the function(s) of all proteins is one of the major challenges in the post genomic area. This long term goal is relevant to fundamental biology as well as translational research such as drug development, since molecular target discovery and validation require an understanding of the function and disease relevance of the proteins. One important step toward the elucidation of their function(s) is to characterize the molecular interacting networks in which they are implicated. In this context, recent modifications of the popular yeast two-hybrid system (Y2H) have been made, enhancing both its stringency and flexibility, resulting in development of new applications such as the reverse yeast two-hybrid system (RY2H) (Bruckner et al., 2009; Vidal & Legrain, 1999).

Y2H is based on the fact that many sequence-specific transcription factors (TFs) increase the rate of transcription of their target genes by binding to cis-acting regulatoring elements (CREs) and activating RNA-polymerase (Pol) II at the corresponding promoters. The DNA-binding and activating functions are usually located in physically separable domains, which are referred to as the DNA-binding domain (DB) and the activation domain (AD), respectively. It has been shown that coupling them respectively with any paired protein–protein interaction domains can reconstitute a functional TF by bringing DB and AD into close physical proximity (Fields & Song, 1989). Thus, the reconstitution of a functional TF can be summarized as DB-X:AD-Y, where X and Y could be essentially any protein domains from any organism. When yeast-growth selection markers such as URA3 or HIS3 (genes involved in uracil and histidine synthesis, respectively) are expressed from a promoter containing DB-binding sites, the DB-X:AD-Y interaction confers a selective advantage. Thus, a few growing yeast colonies can be identified on plates lacking the corresponding amino acid. Such positive selections have been used to identify a great number of specific protein–protein interactions by screening of cDNA libraries (Hamdi & Colas, 2012; James, Halladay & Craig, 1996; Parrish, Gulyas & Finley Jr, 2006).

Conceptually, protein–protein interactions can also be inhibited by the use of cis-acting mutations in one partner (referred to as interaction-defective alleles, IDAs) or trans-acting molecules such as dissociating peptides or small molecules. For example, IDAs can be compared with their wild-type counterparts for their ability to functionally complement a knock-out in the corresponding gene or for their ability to function in an expression assay in the relevant cells. In this reverse version of Y2H, the wild-type DB-X:AD-Y interaction is toxic or lethal for the yeast cells because a toxic marker (e.g., URA3 in presence of 5-fluoroorotic acid, 5-FOA) is used as a reporter gene (negative selection). Yeast cells that express URA3, a gene normally involved in the synthesis of uracil, fail to grow on media containing 5-FOA because the URA3 enzyme transforms 5-FOA into a toxic compound. Thus, DB-X:AD-Y expressing cells are 5-FOA sensitive (5-FOAS). In this setting, DB-X:AD-Y dissociation confers a selective growth advantage that can conveniently identify both interaction-defective alleles. Interactions tested in RY2H can stem from known protein–protein interactions, or from ones inferred computationally (Sardina et al., 2018).

Many different systems and variants have been optimized to reduce the number of false positive clones, resulting in an accumulation of relevant biological data, including data from large scale analysis of the human interactome or protein networks involved in human diseases, in viral replication as well as in drug discovery (Hollingsworth, 2004; Hughes, 2002; Lim et al., 2006; Rual et al., 2005). Consequently, when dealing with large scale analysis coupled to high throughput, Y2H and RY2H generate a huge quantity of positive clones that need to be sequenced in order to identify binding partners and IDAs, respectively. Usually, each positive clone is sequenced and the user gets the results back in one or several fast-all (FASTA) sequence files. When working with large datasets, this step can generate hundreds of such files with the obvious consequence of significantly slowing down the identification of bona fide protein partners. Furthermore, the sequencing product contains part of the cloning vector and a specific primer tag, which might severely limit the efficiency of alignment of clone DNA or protein sequence to public databases via the NCBI Basic Alignment Search Tool (Blast!) for identification (Altschul et al., 1990). An insertion or deletion can also occur during the cloning process, which will disrupt the open reading frame (ORF), resulting into incorrect identification. Finally, conventional sequencing only works well for up to roughly 500–800 base pairs (bp) (Mavromatis et al., 2012). Past that, chromatography signals degrade, thus sequencing becomes less reliable. Therefore, DNA fragments encoding for polypeptides larger than 200 amino acids (aa) cannot be easily sequenced. For RY2H, full length sequencing of such preys is particularly desirable to map the full complement of residue mutations that abrogate a bait-prey interaction. This issue is usually addressed by sequencing both strands using forward and reverse primers. However, this infers subsequent assembly of both sequences and finding the right reading frame in the C-terminus, which is not trivial when dealing with large numbers of positive clones.

Finally, in the case of RY2H, spotting and summarizing bait mutations along the protein primary sequence in order to identify a potential surface of interaction for a given interacting partner requires a dedicated tool. To this end, we developed 2HybridTools, a handy software which simplifies and speeds up these steps by opening several sequencing FASTA files at the same time, by deleting unnecessary vector DNA, by optionally finding the right reading frame, and, in the case of paired-end sequencing, by overlapping corresponding sequencing fragments while trimming low-quality sequencing and keeping the resulting protein in frame, which is then protein-translated and identified via Blast. Additionally, 2HybridTools detects and summarizes mutations in a graphical report, and those mutations can be superimposed onto a protein model sequence if desired. We provide this tool for the users of two-hybrid technology, since there is no freely available such software suite. A commercial suite does exist (DNADynamo). However, it does not support multiple sequences, neither assembly of sequencing products using forward and reverse sequencing. While Contig Assembly Program 3 (CAP3), a DNA assembly program, can be used for this purpose since it supports these features (Huang & Madan, 1999), it is however not dedicated to Y2H and thus lacks features such as identification of clones or trimming of vector sequences. 2HybridTools also improves on these two programs by introducing a model sequence against which clones are aligned, thus producing a summary of specific mutations resulting for example from a RY2H screen, as well as open reading frame detection and automated Blast identification of clones. The efficiency of the program is illustrated here with two examples which we walk through, one from a regular Y2H assay and another to analyze mutations from a RY2H assay.

Methods

Availability

2HybridTools is a portable Java application which requires BioJava (Holland et al., 2008) ≥ 1.5, which is distributed with 2HybridTools and installed in the application’s system classpath, i.e., the lib directory or in the lib/ext directory of the Java runtime environment (JRE) virtual machine folder. Alternatively, BioJava is available at https://biojava.org/. 2HybridTools requires Java ≥ 1.5. The application is entirely open source. Binaries and example files for Y2H as well as RY2H can be obtained freely at GitHub under https://github.com/pc297/2HybridTools.

Algorithm flowchart

An overview of the main steps behind the algorithmic logic used in 2HybridTools is shown as a diagram on Fig. 1. The default mode of processing is Y2H, with no model input. Loading a sequence model enables RY2H mode.

Figure 1 Algorithmic flowchart of the implementation of 2HybridTools.

The flow of information is represented by arrows. Parallelograms represent inputs, rounded rectangles events, straight rectangles processing steps, and diamonds represent conditions. Fields in bold represent mandatory steps.

Main interface

The main interface, a JFrame object, can be viewed in Figs. 2A, 2B, where original sequences are shown (top text pane, a JTextPane object with JScrollBar) and colored according to their sequence nature (i.e., vector, coding sequence, low quality sequence), and processed sequences are aligned with amino acids residues colored according to their physicochemical properties (bottom text pane, a JTextPane object with JScrollBar). Multiple sequence files can be loaded or appended to the current project (Figs. 2A, 2B). DNA sequences are translated into proteins which are aligned and overlapping sequences are then displayed in comparison to sequence model. A File menu allows saving open reading frame analysis, fragment overlapping, translation, Blast and model alignment in either DNA or protein FASTA file formats. Mutation reports can also be saved from the Summary option once alignment to model has been carried out. Multiple options are available (1) vector sequence options (N- and C- Terminus), (2) minimum protein or DNA overlapping size upon N- and C-Terminal sequencing, (3) N-cutoff, i.e., accepted number of unidentified bases before discarding the 3′ remainder of sequences, (4) proxy options for users requiring a proxy for Internet connection (Fig. S1).

Figure 2 Main interface of 2HybridTools in Y2H and RY2H mode.

(A) Functionalities specific to the Y2H mode are highlighted in callout bubbles. These include support for multiple sequences, clone identification via protein Blast and ORF discovery. (B) Functionalities specific to the RY2H mode are highlighted in callout bubbles. These include comparison to a reference sequence, forward and reverse sequencing product assembly and mutation summary.

Sequencing quality trimming

The very start of sequencing files can contain a number of unidentified bases (N) due to poor sequencing quality in the first 15–40 bases at the 5′ end of the fragments. Therefore only the nucleotide sequence following the first N will be used by the program as a default value. Likewise, bases following the second N at the 3′ end will not be used neither. However, if required, the user can modify this default setting (Fig. S1), which brings up a JOptionPane object.

Vector trimming

The default N- and C-Terminus vector tag protein sequences are respectively STHAS and DPAFL, which are tags from pAD vector (Invitrogen™). These sequences can also be entered manually, either as DNA or amino-acid sequences and their position within the sequence will be identified by searching the entire translated sequences in all six open reading frames. This option allows choosing the number of bases in which the protein is in frame downstream of the vector sequence. Therefore, all bases prior to the start of the protein sequence are discarded (Fig. 2A, Fig. S1). To account for possible sequencing errors, mapping of vector sequences can be performed with one mismatch, which will be automatically allowed should the tag sequence not be matched in the prey sequences (Fig. 1). This is achieved by catching a StringIndexOutOfBounds exception when not returning the index of a vector tag sequence in the prey sequence, with a loop going through the entire sequence, matching, in turn, each character of the query, then the next one, until the end of the vector string, with a counter that counts the number of times a character from the query was missed; queries resulting in a counter less than the number of set mismatches are retained. Since the vector sequences are at the beginning of the prey sequences (both for forward and reverse sequencing), we determine those be the first match with a mismatch in the sequence. These options are set via a custom JFrame object with input fields and sliders for text and distance to ORF, respectively.

ORF analysis

When this option is selected, the longest average ORF size will be determined in all sequences. Each sequence is translated in all six ORFs in the selected genetic code. This option can be useful if the fusion protein contains a methionine residue in proximity of the insertion site to correctly identify the frame the insert is in. The strand and frame yielding the highest average translated size will thus be retained. Choosing this option will increase BLAST alignment scores however it should not be used when dealing with fusion proteins. When used, ORF information is added to the sequence identifier.

Protein translation

By default, the universal genetic code is selected. This can however be overridden for use with bacterial two-hybrid, in which case GUG and UUG start codons are documented (Kozak, 1983). ORF and strand are user-defined except in the case of automatic ORF analysis in which these parameters are set up automatically. We made use of the BioJava translation method with the following exception: in the case of a codon containing an N, if the two other bases unambiguously point towards a specific amino acid, e.g., GTN for Valine, then it will be translated as such. Otherwise, the result will be designated as X but will, however, not be treated as a mutation.

Clone identification

HTTP POST requests containing each translated sequence are directed to an Expert Protein Analysis System (ExPASy) (Appel, Bairoch & Hochstrasser, 1994) protein Blast service mirror using plain HTML. HTTP connections are maintained until each query is completed, in which case the HTTP response is parsed and the protein information subsequently retrieved (Fig. 2A). This step is carried out by parsing an InputStreamReader object for hits, with the sequence query written to an OutputStreamWriter object. If no hit was found, this will be reported in the main user interface. Information corresponding to the best hit is added to the sequence identifier. To allow grouping in case of multiple prey proteins, Blast! results can be sorted alphabetically by checking Sort Blast output. Sorting is performed using a Comparator class on BLAST hit information parsed from sequence identifiers.

Sequence overlap for the assembly of forward and reverse sequencing products

Sequences from the same clone are normally labeled similarly, with the name of the primer added, in our example, the N-terminus sequence is labeled AD and the C-terminus labeled “TERM” because we used primers AD and TERM respectively. If an overlap is found between two sequences with the specified minimum overlap, the resulting sequence will be labeled “AD-TERM OVERLAPPED” (Fig. 2B). The minimum overlap is by default set to 6 amino acid residues or 18 bp, corresponding to the minimum number of bases in a complete codon sequence that will be unique in the human genome (16 would be the minimum number of bases however that would yield an incomplete codon). If a corresponding N-terminus is not found for a given C-terminus, the latter will be labeled “TERM NON-OVERLAPPED”. If a C-terminus is not found, neither translation nor overlapping alignment is attempted and the sequence is discarded. The N-terminus is then reverse-complemented from its vector tag onwards. We then use a local pair-wise Smith-Waterman alignment (Smith & Waterman, 1981) at the amino acid or DNA level with high substitution, insertion, deletion and gap penalties, using an International Union of Pure and Applied Chemistry (IUPAC) identity substitution matrix. The largest possible alignment with the best score is used in order to overlap N- and C-termini. A minimum alignment length of 5 perfect matches is required for N- and C- terminus overlap, although this can be set as a parameter. This step is carried out by calling a SmithWaterman object from BioJava.

Sequence model alignment and mutation report

For RY2H, a DNA or protein model sequence can be loaded prior to analysis, which will result in all translated sequences being aligned in turn to the model protein sequence. The model can either be loaded from a local FASTA file, or downloaded directly from NCBI by inputting the protein accession number. This is achieved by posting a URL containing the parsed NCBI accession number, eliciting a response in plain FASTA text format. Selecting a model following translation will not cause sequences to be translated or overlapped again. We perform a Smith-Waterman alignment (by calling a SmithWaterman object from BioJava) with low substitution penalties and high insertion, deletion and gap penalties using the BLOcks SUbstitution Matrix clustered at the 62% level (BLOSUM62) substitution matrix (Henikoff & Henikoff, 1992). Non-overlapped C-terminal fragments are also aligned. Only alignment scores greater than the specified or default alignment score, which corresponds to 5 × model sequence length × 80%, are kept. Sequences with alignment scores satisfying these parameters are maintained in the lower text pane (Fig. 2B). To limit the rate of false positive discovery, including frame shift and termination mutations, sequences shorter or longer than the sequence model are discarded when retrieving mutations within an alignment. Results are then summarized above the sequence of reference with the precise mutation nature and counts and can be viewed by clicking the Summary button (Figs. 3A, 3B). Finally, another graphical, color-coded overview of the mutation distribution along the protein can be obtained by pressing the Graph button (Fig. 3C).

Figure 3 Mutation summaries and localization using a reference sequence in RY2H mode.

(A, B) Mutation reports of Nsp16 clones disrupting interaction with Nsp16, 1-50 aa (A) and 51-100 aa (B). (C) Graphical overview of mutation location on Nsp16, with the number of mutations indicated as a color-coded heatmap, from left to right.

Comparison to other software

As mentioned above, other solutions that encompass sequence assembly already exist, and can thus be readily used for Y2H analysis. A summary of the functionalities of 2HybridTools compared to other similar programs is shown on Table 1. CAP3 (Huang & Madan, 1999), a recognized all-purpose sequence re-assembly program and one of the most widely used tool, can be used for the purpose of clone re-construction as it implements sequencing quality cut-off, clone identification using the latter. But it cannot assemble sequence(s) containing vector sequences. CAP3 was not written specifically for the purpose of Y2H/RY2H analysis. Conversely, 2HybridTools features vector-trimming and multiple sequence Blast, which greatly facilitates clone identification. Moreover, CAP3 does not implement protein translation, which renders protein Blast impossible and thus would incorrectly align a DNA sequence containing a deletion induced by induced or spontaneous mutagenesis to a reference sequence. DNADynamo, a commercial software suite, also supports several of the above mentioned functions but does not support multiple sequences. Furthermore, 2HybridTools provides a unique, RY2H-dedicated mutation report function, which allows pinpointing mutations that could define a surface of interaction that plays an important role in protein–protein interactions. Overall, implementation of the automated Blast, vector trimming and translation features provided in 2HybridTools allow easier clone identification compare to CAP3, and provides a unique, RY2H-dedicated mutation report function, which is absent in DNADynamo. This feature thus allows an in-depth Y2H/RY2H analysis, as detailed below.

Table 1 Features supported by 2HybridTools compared to available programs used for Y2H.

Features/Program	2HybridTools	CAP3	DNADynamo	
Assemble forward and reverse sequences	✓	✓	✓	
Multiple sequence support	✓	✓	×	
Vector trimming	✓	×	✓	
Vector library	×	×	✓	
Protein translation	✓	×	✓	
Mutation summary	✓	×	×	
Integrated Blast	✓	×	✓	
Open reading frame identification	✓	×	✓	
Map restriction sites	×	×	✓	
Free licence	✓	✓	×	

Results

SP1 Y2H Screening

A yeast two-hybrid screening of the transcription factor SP1 (Kadonaga et al., 1987) was performed at the two-hybrid platform, CRCM (Inserm UMR891, Marseille, France) using a previously described protocol (Walhout & Vidal, 2001). The screen was performed using a human colon cDNA library, in order to discover potential new partners for SP1 involved in apoptosis (Vicart et al., 2006). We read the resulting FASTA sequencing files corresponding to positive clones (“SP1_2hybrid_screening.seq”) into 2HybridTools using the File/Open Sequence(s) drop-down menu (Fig. S3A). For this particular experiment, the vector tag sequence was STHAS, which we set in the Vector/Define 5′ Vector menu (Fig. S3B), with all other options left unchanged from their default values. ORF analysis in 2HybridTools, performed by pressing the Find ORFs button, correctly identified the first codon encoded by the prey and the right reading frame in all the clones isolated from the screen. This identification was necessary since several clones encoding the same partner were identified but they were not necessarily starting at the same amino acid within the protein. This is due to the cloning strategy used to create the cDNA library. To translate cDNA into protein and to trim vector sequences, as well as low sequencing quality sequencing ends, we hit the Go button, updating the main interface with the resulting trimmed sequences and translated sequences (Fig. 2A, top and bottom text fields). We then set out to identify the protein sequences of positive clones using the built-in automated Blast! feature, pressing the Blast results button in the main interface. This analysis revealed that the proteins CDX1 and RPL23 were overrepresented in the sequencing results (Fig. 2A), yielding respectively 20 and 3 hits each. These proteins are thus likely partners of SP1. SP1 is known to interact with homeobox proteins (Park, Kim & Han, 2007) and furthermore CDX1 is a direct transcriptional target of SP1 (Gilmour et al., 2014; Lim & Chang, 2009). Interaction with RPL23 suggests that this interaction is cytoplasmic and could involve the synthesis or transport processes of the SP1 protein.

Nsp10-Nsp16 Reverse Y2H Screening

We have also performed a RY2H screening using the SARS non-structural proteins (Nsp) Nsp10 and Nsp16 (Imbert et al., 2008; Lugari et al., 2010). Nsp10 was used as the bait to identify interaction-defective alleles (IDAs) of Nsp16. To this end, we generated a library of randomly mutated full-length Nsp16 that was screened against Nsp10 (Lugari et al., 2010). The Nsp16 cDNA being 897 bp long, we sequenced positive clones (whereby the interaction between the bait and prey is abrogated) from both ends. This cDNA length allows forward and reverse sequencing products to overlap. As with conventional Y2H, we input the resulting FASTA sequencing files (MultiFastaNsp16 mutants.seq, containing both forward and reverse sequencing products) using the File/Open Sequence(s) menu (Fig. S3C). We also read the Nsp16 reference cDNA sequence (NSP16DNAmodel.txt) using the File/Load Model (Optional) menu (or, alternatively, as a protein sequence, using the file NSP16PROTEINmodel.txt) (Fig. S3C). For this experiment, we set the vector tag sequence as STHAS and the field “Distance to ORF”, the distance of the vector tag sequence, to the ORF as 6 amino acids (Vector/Define 5′ Vector) (Fig. S3D, top). Since positive clones were sequenced from both ends, we also input a 3′ vector tag sequence as DPAFL (Vector/Define 3′ Vector), leaving the field Distance to ORF to 0 as the ORF ends directly upstream of this sequence (Fig. S3D, bottom). We set all other parameters to their default values. We then hit the Go button in the main interface, starting processing of sequences, overlapping of sequencing ends, and aligning to the reference sequence, updating the main display with trimmed sequences (Fig. 2B). This analysis allowed reconstructing 91 Nsp16 full-length proteins (from 182 DNA sequences) out of 189 original N- and C-Terminus cDNA sequences (Fig. 2B). Only 7 sequences were not overlapping, of which 4 correspond to non-matched N-Terminus fragments. To obtain a detailed per-residue summary of mutations in the cDNA sequences corresponding to positive prey proteins, we hit the Summary button in the main interface. A detailed view of the mutations on the first 100 residues of Nsp16 that led to lost Nsp10-Nsp16 is shown on Figs. 3A, 3B. To bring up a heatmap view of mutations, we hit the Graph button in the main interface. A summary of the residue mutation frequencies on the entire Nsp16 protein is shown on Fig. 3C.

Discussion

2HybridTools improves on other non-dedicated software used for Y2H analysis software by combining features uniquely supported by some in an all-in-one fashion. To our knowledge, our software is the first to implement dedicated RY2H analysis. The alignment to a reference sequence as well as the identification of mutation positions, nature and frequencies provides a valuable tool for RY2H. Ensuring that positive sequences, which resulted in abolishing the bait/prey interaction, were in frame also allows dismissing false positives. The ability to align sequencing products from both ends when dealing with sequences longer than 500–800 bp in an automated fashion, for multiple sequences, also provides a net gain of analysis time for Y2H and RY2H end-users, as does the automated identification of multiple clones. Thanks to its Java implementation, 2HybridTools is also platform-independent and thus easily accessible to Y2H/RY2H users. The ability to identify specific mutations from RY2H also provides the basis for focused study of interaction residues. Although only more direct protein–protein interaction assays such as glutathione S-transferase (GST) pull-down, bioluminescence or fluorescence resonance energy transfer (BRET or FRET) (Lohse, Nuber & Hoffmann, 2012) will yield definitive results as was done for Nsp10 and Nsp14 (Bouvet et al., 2014), the combined use of this tool and 3D structures is a good indicator as to where to start looking using these high-resolution techniques.

Conclusions

A possible future development for this tool would be the automated mapping of mutations using 3D modeling software such as Rasmol and automatically generated Rasmol scripts, which would directly map mutations on a given crystal structure. In fine, our software can be used to expand current protein–protein interaction databases. Finally, as recent developments to Y2H/RY2H have been made to exploit next-generation sequencing (NGS) to sequence positive clones (Suter et al., 2015), a further development to this software would entail the use of NGS sequencing results.

Supplemental Information

Figure S1 Options of 2HybridTools

Drop down menus showing (A) file options, (B) vector options: 5’ and 3’ vector sequences, mismatches and (C) overlap options for assembly of forward and reverse sequencing products.

Click here for additional data file.

Figure S2 Optional sorting of Blast! output in 2HybridTools

Alphabetical sorting is enabled by the Sort Blast output checkbox.

Click here for additional data file.

Figure S3 Walkthrough of Y2H and RY2H examples

(A, B) Y2H mode, loading sequences (A), and setting 5’ vector tag sequence as STHAS, with ORF starting directly after tag sequence (B). (C,D) RY2H mode, loading sequences as well as reference model (C), and setting 5’ vector tag sequence as STHAS, with ORF starting 6 residues after tag sequence (D, top); setting 3’ vector tag as DPAFL, with ORF starting ending directly before tag sequence (D, bottom).

Click here for additional data file.

We express our special thanks to Frédéric Rosa for original testing.

Additional Information and Declarations

Competing Interests

Author Contributions

Data Availability

The authors declare there are no competing interests.

Pierre Cauchy conceived and designed the experiments, analyzed the data, contributed reagents/materials/analysis tools, prepared figures and/or tables, authored or reviewed drafts of the paper, approved the final draft, wrote all computer code for implementation.

Brigitte Kahn-Perlès conceived and designed the experiments, performed the experiments, contributed reagents/materials/analysis tools, approved the final draft.

Pierre Ferrier, Jean Imbert contributed reagents/materials/analysis tools, prepared figures and/or tables, authored or reviewed drafts of the paper, approved the final draft.

Patrick Lécine conceived and designed the experiments, performed the experiments, analyzed the data, contributed reagents/materials/analysis tools, authored or reviewed drafts of the paper, approved the final draft.

The following information was supplied regarding data availability:

pc297/2HybridTools Data is available at GitHub: https://github.com/pc297/2HybridTools.

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
