# Peer review of "HybridTools, a handy software to facilitate clone identification and mutation mapping from yeast two-hybrid screening"

_PeerJ, doi:10.7717/peerj.7245_

## Round 0.1 · original submission · Minor Revisions

Both reviewers have recommended to incorporate some changes to your manuscript. I agree with the reviewers that these changes will improve your manuscript. Especially, instructions on installing BioJava and installation and usage of Github. In addition, both reviewers recommended some additional changes that should be addressed in the revision.

Reviewer 1 ·

Basic reporting

no comment

Experimental design

no comment

Validity of the findings

no comment

Additional comments

The manuscript introduces a convenient software to facilitate Y2H clone identification and mutation mapping, namely 2HyridTools. Though related software with partial similar functions exist, the authors made a good justification for the necessity of a dedicated Y2H software. The implementation and the test results were clearly stated. This tool will be useful to many users in the future. It is recommended to be published in PeerJ.

There are just four minor comments.

1.The program depends on BioJava. It would be better if it is independent, or detailed instruction should also provide for how to install BioJava.

2.Lines 158 – 161, does vector trimming allow mutations on the tag sequences, which may arrive from sequencing errors?

3.Lines 192 – 193, please clarify “If a C-terminus is not found, only the C-termnus will be translated, but not overlapped…”

4.How does the software handle the scenario that within a single experiment, different sequences correspond to different prey proteins interacting with the same bait proteins? Will it group these sequences according to prey proteins?

Reviewer 2 ·

Basic reporting

In the manuscript “2HybridTools, a handy software to facilitate clone 1 identification and mutation mapping from yeast two-hybrid screening”, the authors present a new tool for the post-processing analysis for experiments using yeast two-hybrid (Y2H) and reverse two-hybrid (RY2H).
The manuscript conforms to the structure recommended by the journal viz. Introduction, Materials and Methods, Results, Discussion, Conclusions. There is continuity in the flow for the readers. I appreciate that the authors have provided references wherever applicable. I appreciate the author for providing the source code and making the software publicly available. The English and grammar of the article are good with very few typos. Below are points that might be useful to further improve the paper:

1. The section “Implementation” should be renamed as “Methods”
2. Provide a documentation for installation and usage of the software on the Github page
3. Provide full-form at the first occurrence of the abbreviation
4. Provide a diagram highlighting the functionalities of the software and the method used to achieve it. I would suggest making it as figure 1.
5. It would be very useful for the readers and make the software easily adopted by researchers, if you can walk-through an example Y2H and RY2H experiment.

Experimental design

The experimental design has no flaws.

Validity of the findings

The tool is an important contribution for researchers working with Y2H and RY2H. Further improvements as suggested in Basic Reporting section would be useful to the community.

---

## Round 0.2 · accepted · Accept

This will be a valuable software for the scientific community.